# Isolation and Characterization of Yeast with Benzenemethanethiol Synthesis Ability Isolated from Baijiu Daqu

**DOI:** 10.3390/foods12132464

**Published:** 2023-06-23

**Authors:** Guihu Zhang, Peng Xiao, Youqiang Xu, Honghua Li, Hehe Li, Jinyuan Sun, Baoguo Sun

**Affiliations:** 1Key Laboratory of Geriatric Nutrition and Health, Beijing Technology and Business University, Ministry of Education, Beijing 100048, China; zhanggh0634@163.com (G.Z.);; 2China Food Flavor and Nutrition Health Innovation Center, Beijing Technology and Business University, Beijing 100048, China; 3Key Laboratory of Brewing Molecular Engineering of China Light Industry, Beijing Technology and Business University, Beijing 100048, China

**Keywords:** *Saccharomyces cerevisiae*, baijiu, daqu, benzenemethanethiol, formation mechanism

## Abstract

Baijiu, a prevalent alcoholic beverage, boasts over 2000 aroma compounds, with sulfur-containing compounds being the most influential in shaping its flavor. Benzenemethanethiol, a distinctive odorant in baijiu, is known to enhance the holistic flavor profile of baijiu. Despite its importance, there is very little literature on the biotransformation mechanism of benzenemethanethiol. Thus, extensive research efforts have been made to elucidate the formation mechanism of this compound in order to improve baijiu production. In this study, 12 yeast strains capable of generating benzenemethanethiol were isolated from baijiu daqu, and the *Saccharomyces cerevisiae* strain J14 was selected for further investigation. The fermentation conditions were optimized, and it was found that the optimal conditions for producing benzenemethanethiol were at 28 °C for 24 h with a 4% (*v/v*) inoculum of 3.025 g/L L-cysteine. This is the first time that yeast has been shown to produce benzenemethanethiol isolated from the baijiu fermentation system. These findings also suggest that benzenemethanethiol can be metabolized by yeast using L-cysteine and benzaldehyde as precursor substrates.

## 1. Introduction

Baijiu, a traditional alcoholic beverage crafted from cereals, undergoes a solid-state spontaneous fermentation process in an open environment. Generally, the raw materials, manufacturing process, tools, and environmental variables for the production of baijiu exhibit notable distinctions from those utilized in the distillation of spirits such as whisky. These disparities contribute to the creation of a distinctive flavor profile that sets baijiu apart from other distilled liquors. During the manufacturing of baijiu, jiuqu (a saccharifying and fermenting agent) is mixed with materials to facilitate simultaneous saccharification and fermentation. Following the fermentation stage, the mixture undergoes solid-state distillation to yield the base baijiu. Subsequently, the base baijiu is meticulously stored in stainless steel containers or pottery jars, with subsequent blending conducted in a sequential manner [1]. This process creates a complex and delicately balanced micro-ecological environment consisting of microbiota, including yeasts, molds, and bacteria; it is within this milieu that the remarkable and distinct aromatic properties of Baijiu emerge, embodying a captivating tapestry of flavors [1,2]. With twelve distinct types of baijiu, the manufacturing process and aroma characteristics play a crucial role in determining the distinction and quality of baijiu [3,4]. Aroma dominates the flavor profile of baijiu, as it is abundantly enriched with a plethora of aroma compounds, owing to its intricate and elaborate production process. Ethanol and water account for 98% of the total weight of baijiu, while flavor compounds make up the remaining 2% [5]. Baijiu is known to contain in excess of 2000 aroma odorants, encompassing a diverse range of compounds such as alcohols, esters, aldehydes, acids, and sulfur compounds, which contribute to its distinct aroma [6,7]. Sulfur-containing compounds are particularly noteworthy as they have an extremely low threshold, low concentration, and distinctive aroma characteristics. They can enhance the overall aroma of baijiu at trace concentrations, high or incongruous levels can lead to off-odor [8,9,10]. Several compounds, such as 2-furfurylthiol and benzenemethanethiol, have been identified in baijiu and are known to promote its sesame, roasted, and overall flavor and quality [11,12,13]. Notably, benzenemethanethiol has garnered recognition as a significant aroma compound within the flavor spectrum of baijiu, owing to its profound presence and substantial impact on the sensory experience of baijiu [11,12,14]. Despite a trace concentration range of 0.02 μg/L–90 μg/L, a conclusion has been substantiated through the implementation of aroma recombination and omission tests [13,14,15].

Li and colleagues have identified benzenemethanethiol as a pivotal contributor to the aroma profile of Chinese sesame-flavored baijiu [12]. Since then, several studies have quantified the levels of benzenemethanethiol in various baijiu samples using different analytical methods, with trace amounts detected at 4.87 ± 0.27 μg/L and an odor activity value (OAV) of 487 signifying its substantial flavor contribution [11,13,14,15]. However, despite these efforts, the mechanism behind the formation of benzenemethanethiol during baijiu fermentation remains enigmatic. Previous studies have suggested that yeast may transform cysteine–benzaldehyde adducts into benzenemethanethiol through the action of β-(carbon–sulfur) lyase enzyme [16,17,18,19] (Figure 1). Hence, our hypothesis suggests that yeast may employ L-cysteine and benzaldehyde as substrates, leading to the production of benzenemethanethiol through biochemical reactions. This knowledge gap underscores the urgent need for further exploration and research in this particular area.

The main objectives of this study were to (i) isolate and identify yeast species that can synthesize benzenemethanethiol from the baijiu fermentation system, and (ii) confirm the mechanism of benzenemethanethiol formation in the yeast isolates described in previous research.

## 2. Materials and Methods

### 2.1. Materials and Chemicals

All chemicals used in this study were of high purity, with a minimum purity level exceeding 95%. L-cysteine was acquired from Biorigin (Beijing, China), benzaldehyde was obtained from Aladdin (Beijing, China), diisopropyl disulfide was purchased from MREDA Scientific Co., Ltd. (Beijing, China). Benzenemethanethiol was purchased from 9 Ding Chemistry Co., Ltd. (Shanghai, China). Ultrapure water was prepared by using the Milli-Q system. The four different types of extract fibers were obtained from Supelco (Bellefonte, PA, USA).

### 2.2. Instruments and Equipment

The main instruments required for the experiment are listed in Table 1.

### 2.3. Sample Collection

The daqu samples were sourced from 3 sauce-aroma-type baijiu distilleries. Five bricks of each off-room daqu were randomly picked from a fresh batch of fermentation workshops. The bricks were pulverized to allow for sieving mesh. Daqu powders were preserved at 4 °C until required.

### 2.4. Yeast Isolation and Identification

To isolate yeast, 10 g of samples was suspended in 90 mL of sterile ultrapure water and shaken at 200 rpm for 30 min under ambient temperature environment. The suspensions were then serially diluted (10^−1^–10^−6^) and plated in duplicate on Yeast Peptone Dextrose (YPD) agar plates and cultivated for 48 h at 30 °C. Subsequently, a single colony was carefully chosen based on variations in colony color, morphology, and frequency for the purpose of separation and purification. Subsequent fermentation tests and identification were performed.

#### 2.4.1. Polymerase Chain Reaction Amplification of 5.8S rDNA

Yeast was cultured into YPD medium, followed by harvesting the sample through centrifugation. A PCR system was used to amplify the genomic DNA of the sample extracted by using a Fungus DNA Extraction kit. For amplification of the ITS1/ITS4 domain of the 5.8S rDNA region, primers 1 (5′- TCCGTAGGTGAACCTGCGG-3′) and 2 (5′- TCCTCCGCTTATTGATATGC-3′) were used [21]. The PCR reaction system and conditions adhered to the manufacturer’s specifications, as outlined in Table 2 and Table 3.

#### 2.4.2. Purification and Sequencing of PCR Product

The gel-cutting purification sequencing process was conducted following the instructions provided by the manufacturer. Sequence similarity alignment was then performed using BLAST (Version 2.14.0; Beijing Meiyounuo Biotechnology Co., Ltd.; Beijing, China) against the NCBI GenBank database. ITS1/ITS4 regions of the 5.8S rDNA region representing the same species shared >98% nucleotide identity.

### 2.5. Fermentation Conditions

In accordance with a previous report, the fermentation medium was prepared with slight modifications [21]. A brief overview of the process involves mixing 1 kg of ground sorghum powder with 4 L of deionized water, followed by stewing for 2 h. Subsequently, the mixture underwent saccharification at 60 °C for 4 h with glucoamylase (50 U/mL). Furthermore, the resulting supernatant was collected by using gauze filtration and centrifugation at 8000 r/min for 10 min, yielding the sorghum extract medium. Leica refractometer (Fisher Scientific, Pittsburgh, PA, USA) was used to measure sugar content; the final sugar concentration was diluted to 7 °Bx with water. Each 100 mL of sorghum extract was transferred to 250 mL Erlenmeyer flasks and sterilized at 121 °C for 15 min prior to use [22].

Furthermore, the fermentation medium contained 1.06 g/L of benzaldehyde and 1.21 g/L of L-cysteine as substrates for benzenemethanethiol. Both benzaldehyde and L-cysteine underwent purification through filtration and were mixed with the sterile sorghum extract medium. This final mixture served as the liquid fermenting media. The yeast strain was pre-cultured in YPD liquid medium at 30 °C, 200 r/min for 24 h. Then, 250 mL Erlenmeyer flasks containing 100 mL of sterile liquid fermentation medium were inoculated with 2% (*v*/*v*) of the yeast. Stationary fermentations were performed at 30 °C for 48 h, and an uninoculated sample of the fermentation medium acted as control.

### 2.6. Quantitation of Benzenemethanethiol in the Fermentation Broth

#### 2.6.1. Optimization of Extraction Condition

For the selection of SPME fibers, this study investigated four different coated fibers: divinylbenzene/carboxen/polydimethylsiloxane, 50/30 μm (DVB/CAR/PDMS); carboxen/polydimethylsiloxane, 75 μm (CAR/PDMS); polydimethylsiloxane, 100 μm (PDMS); polyacrylate, 85 μm (PA). These fibers were procured from Supelco (Bellefonte, Pa., USA), were 1 cm in length, and underwent conditioning according to the manufacturer’s specific protocol. To optimize the HS-SPME process, a certain amount of benzenemethanethiol was added to a 20 mL glass vial containing 8 mL sorghum extract medium to ensure matrix homogeneity. Additionally, 2 g of NaCl and a 6 mm Teflon-coated stirring rod were added to the glass vial. The sample was then equilibrated at 30 °C in a water bath for 30 min under stirring at 250 rpm. Following equilibrating, the SPME fiber was inserted into the headspace of the vial for 30 min at 30 °C. After the extraction step, the fiber was penetrated into the GC injection port for desorption at 250 °C for 7 min.

#### 2.6.2. HS-SPME Parameter Optimization

The HS-SPME conditions were adjusted to optimize the extraction of benzenemethanethiol for subsequent detection. The names and levels of variables were chosen for optimization of the response variable, which are as follows: salt dosage (0 g/8 mL, 1 g/8 mL, 2 g/8 mL, 3 g/8 mL, and 4 g/8 mL), extraction time (t_ext_: 10, 20, 30, 40, and 50 min), extraction temperature (Text: 25, 30, 35, 40, and 45 °C), desorption time (t_d_: 1, 3, 5, 7, and 9 min).

#### 2.6.3. GC-SCD Analysis

For the analysis of benzenemethanethiol generated by different yeast strains during fermentation, gas chromatography–sulfur chemiluminescence detection (GC-SCD) was employed. After the fermentation process was complete, 10 mL of the sample was sampled and centrifuged at 8000 r/min for 10 min at 4 °C before analysis. The supernatant was then harvested and preserved at −20 °C. Next, 8 mL of the supernatant was transferred into a 20 mL headspace vial containing 3 g of sodium chloride. A 75 μm (CAR/PDMS) fiber was used to absorb the target compounds, and the sample was equilibrated for 30 min and extracted at 30 °C for 30 min. After extraction, to allow for desorption of the analytes via the split-less mode, the fiber was inserted into the injection port of the GC (250 °C) for 7 min.

GC-SCD analysis of benzenemethanethiol was conducted on an Agilent 7890A GC (Agilent Technologies Inc., Santa Clara, CA, USA) and an SCD detector 8355. The instrument was equipped with a DB-WAX capillary column (30 m × 0.25 mm i.d., 0.25 μm film thickness). High-purity nitrogen served as carrier gas at a flow rate of 1 mL/min. The oven temperature program was as follows: the temperature was maintained at 50 °C for 2 min, increased to 150 °C at a speed of 3 °C/min, raised to 230 °C at 5 °C/min, and maintained for another 10 min [13]. For the Agilent 8355 SCD, the burner temperature is set at 800 °C, the base temperature is set at 200 °C, oxidant flow and hydrogen flow are set at 50 and 46, respectively. Chemical identification was conducted using a comparison of the retention times of target compounds on the columns with the authentic standards. Semi-quantification was conducted in line with the previous articles published by research groups [9,22]. Diisopropyl disulfide served as the internal standard (IS) and used at a final concentration of 1.35 mg/L in samples.

### 2.7. Optimization of Benzenemethanethiol Production Conditions

Sulfur-containing compounds possess the intriguing quality of making a “Low Quantity but Critical Contribution to Flavor”. However, within the realm of Baijiu microbiology, there is a dearth of research concerning microorganisms that are proficient in generating sulfur-containing compounds. Previous studies have revealed that microorganisms produce these compounds in limited quantities due to inadequate understanding of their metabolic mechanisms. Consequently, the objective of the current study is to identify factors that potentially influence the production of benzenemethanethiol and investigate their impact on fermentation performance using single-factor experiments. In the present study, four conditions were optimized to obtain better enzyme production efficiency. The isolated and purified strains were inoculated into the fermentation medium with the following inoculum size (2% *v/v*). The optimized parameters were inoculation volume (1%, 2%, 3%, 4%, 5% *v/v*), amount of L-cysteine addition (0.605, 1.21, 1.815, 2.42, 3.025, 3.63 g/L), fermentation temperature (20, 25, 28, 30, and 35 °C), and fermentation time (6, 12, 24, 36, 48 h). For inoculation volume, the amount of L-cysteine addition, fermentation temperature, the concentration level of benzenemethanethiol in the fermentation broth were measured after two days of fermentation. The sampling time during the fermentation process was set according to the previously literature reported by our groups [20]. Given that the composition of the fermentation medium remains consistent, the concentration of benzenemethanethiol in the sample can serve as an indirect measure of the carbon–sulfur lyase efficiency of the functional microorganism. The highest benzenemethanethiol content signifies the optimal carbon–sulfur lyase production efficiency of *Saccharomyces cerevisiae*.

## 3. Results and Discussions

### 3.1. Optimization of HS-SPME Extraction Conditions

The selection of fiber coatings is crucial for achieving optimal adsorption efficiency. Prior to analyzing benzenemethanethiol, we optimized parameters to enhance extraction efficacy. The efficiency of solid-phase microextraction (SPME) fibers can vary depending on the coating material used [23,24]. Several studies have employed various extraction fibers to characterize sulfur-containing compounds in foods and beverages [13,15,25,26,27]. In this study, we tested four types of SPME fibers (DVB/CAR/PDMS, CAR/PDMS, PDMS, and PA) for their ability to extract benzenemethanethiol from sorghum extract. Fibers with DVB/CAR/PDMS and CAR/PDMS coatings demonstrated superior capacity for extracting benzenemethanethiol compared to the other fibers (Figure 2). Specifically, the response areas for benzenemethanethiol using the CAR/PDMS fiber were significantly better than the other three fibers, with a relative standard deviation (RSD) of less than 10% (*p* < 0.05). Our findings are consistent with a recent study that used a CAR/PDMS fiber coupled with GC × GC–SCD to extract volatile sulfur-containing odorants in Chinese Laobaigan baijiu [13]. Therefore, CAR/PDMS was chosen for the next period of HS-SPME optimization in fibers test under equivalent conditions, as it manifested the optimal extraction efficiency and was excellent for extracting benzenemethanethiol from the sorghum extract medium.

The optimization of Solid Phase Microextraction (SPME) conditions for extracting benzenemethanethiol were carried out using a single-factor approach, focusing on salt dosage, extraction time and temperature, and desorption time as independent variables. The addition of NaCl was found to have a stimulating effect on the release of volatile compounds in food matrices by increasing the ionic strength of the sample and diminishing the solubility of analytes. This, in turn, increased the distribution of corresponding compounds on the fiber coating and enhanced the adsorption of volatile substances by the extraction fiber. This study investigated the effects of five different levels of salt addition on the extraction performance of benzenemethanethiol. The peak area of benzenemethanethiol demonstrated an initial increase, followed by a decrease, with increasing addition of sodium chloride, reaching the highest level at 3 g/8 mL. Therefore, 3 g/8 mL was selected as the optimal salt dosage for further HS-SPME optimization.

The attainment of equilibrium within the system is influenced by an array of factors, encompassing the distribution coefficient of the components, the rate of substance diffusion, the characteristics of the sample matrix, the volume of the sample, and the thickness of the fiber membrane employed for extraction [28]. Figure 3 demonstrates that the chromatographic peak of benzenemethanethiol exhibited a gradual augmentation as the duration of extraction was prolonged, albeit with a diminishing rate of increase. Moreover, when the extraction time exceeded 30 min, there was no significant difference in the extraction efficiency. Consequently, 30 min was identified as the optimal extraction time.

Elevating the temperature during extraction enhanced the distribution coefficient and accelerated analyte mass transfer, ultimately resulting in improved extraction efficiency. Within a temperature range of 25–45 °C, the amount of extracted benzenemethanethiol exhibited a positive correlation with increasing temperature. However, beyond 30 °C, no significant difference in extraction efficiency was observed. As the temperature continued to increase, stray peaks appeared on the chromatogram, possibly due to spontaneous reactions of benzenemethanethiol producing other compounds, as observed in previous studies on thiols subjected to temperature-related staling issues [29,30]. Hence, 30 °C was established as the optimal temperature for subsequent analyses.

The duration of desorption exerted a significant impact on the content of the desorbed substances. Insufficient desorption time may lead to the incomplete release of volatile elements, whereas excessive desorption time can result in a notable decrease in the longevity of the extraction fiber. Hence, an investigation was conducted to evaluate the influence of various desorption times on the detection outcomes of benzenemethanethiol. The findings, as depicted in Figure 3, demonstrate that the optimal resolution time was 7 min.

In conclusion, we have determined that the ideal pretreatment conditions involve using a 75 μm (CAR/PDMS) extraction fiber, a 30 min equilibration period, an extraction temperature of 30 °C, a 30 min extraction duration, the addition of 3 g of salt, and 7 min of desorption.

### 3.2. Determination of Benzenemethanethiol Biosynthesis in Yeast Isolates

Daqu not only provides materials for baijiu fermentation, but also enriches the fermentation environment with a diverse range of functional microbial communities that act as driving forces during manufacturing [31,32,33]. These microbes in the fermentation environment significantly facilitate the generation of flavor profiles of baijiu. In this study, we identified a total of 39 yeast strains segregated from Daqu, with most of the strains being preliminarily determined as *Saccharomyces cerevisiae* based on morphological characteristics. Previous studies have demonstrated that *S. cerevisiae* is the most common yeast species in baijiu fermentation and is the primary contributor to the production of ethanol, higher alcohols, and esters [3,34,35]. Benzenemethanethiol is a unique aroma compound that enhances the aroma of several types of baijiu, such as soy-sauce-aroma-type baijiu, strong-aroma-type baijiu, roasted-sesame-like-aroma-type baijiu, and fuyu-aroma-type baijiu [11,15,25]. Therefore, it becomes imperative to identify the specific yeast species or strains capable of biosynthesizing benzenemethanethiol to enable informed strategies for enhancing baijiu fermentation in a targeted manner.

The biosynthesis of benzenemethanethiol was assessed across all yeast strains. GC-SCD analysis (Appendix A) confirmed that 12 yeasts could produce benzenemethanethiol using L-cysteine and benzaldehyde as precursors. Notably, the ability to synthesize benzenemethanethiol varied among the strains, consistent with the findings reported by [21]. Yeast J14 exhibited a notably higher production capacity for benzenemethanethiol compared to the other strains (Table 4), leading to its selection for further investigation in the subsequent study.

### 3.3. Identification of High Carbon–Sulfur-Lyase-Producing Strains

#### 3.3.1. Electrophoresis of PCR Product

In this study, PCR was used to amplify the 5.8S rDNA using universal J14 primers (ITS1 primer and ITS4R primer) (Appendix A). The bands observed were unambiguous and clear without any specific amplification, and the PCR products were approximately 800 bp in size (Appendix A). These electrophoresis results confirmed the credibility of the PCR amplification and enabled appropriate subsequent sequencing. In addition, we extracted the genomes of the remaining 11 yeast strains capable of producing benzyl mercaptan and used ITS sequences to perform PCR amplification, obtaining corresponding photos of genome extraction (Appendix A).

#### 3.3.2. Construction of the Evolutionary Tree

The 5.8S rDNA sequence of J14 was compared to the corresponding reference gene sequence in the National Center for Biotechnology Information (NCBI) sequence database (BLAST). The results revealed that the 5.8S rDNA of J14 demonstrated over 99% homology with the genus *Saccharomyces cerevisiae*. The construction of the phylogenetic evolutionary tree is presented in Figure 4. By examining the nodes among each generation in Figure 4, it is possible to infer the most closely related microbial genera and determine the degree of intergeneric matching through node scores. The position of the “J14” marker in Figure 4 indicates the genus information for J14.The strains with the highest degree of similarity to J14 were identified as *Saccharomyces cerevisiae*, with over 95% homology.

### 3.4. Optimization of Fermentation Condition

#### 3.4.1. Effect of Seed Volume

To ensure a consistent and optimal production level of benzenemethanethiol, we investigated the impact of seed volume on yield. Typically, increasing the amount of inoculum or introducing seed liquid during the logarithmic growth phase is employed to minimize the lag period and enhance fermentation efficiency. In this study, we examined the effects of different inoculation volumes, specifically 1%, 2%, 3%, 4%, and 5% (*v/v*), on the fermentation process, as depicted in Figure 5A. Overall, no significant differences were observed among the different inoculation volumes during the fermentation process. Nevertheless, the benzenemethanethiol content initially decreased and then increased with increasing inoculation amounts, with a maximum yield of 0.25 mg/L achieved at an inoculation amount of 5% (*v/v*). It should be noted that there was a significant fluctuation in output at each inoculation amount level, which may be attributed to yeast toxicity caused by the addition of benzaldehyde or other factors. Considering production stability and yield, we selected 4% (*v/v*) of the inoculation volume for subsequent fermentation condition optimization.

#### 3.4.2. Effect of L-Cysteine Supplementation Amount

Sulfur amino acids play a crucial role in the formation of volatile sulfur compounds, such as 4-mercapto-4-methylpentan-2-one (4MMP), 3-mercaptohexan-1-ol (3MH), and 3-(methylthio)-1-propanol [36,37,38,39]. Previous research has demonstrated that L-cysteine is a core substrate for 2-furfurylthiol, and its yield is positively correlated with L-cysteine concentration [40,41,42]. Therefore, we hypothesized that the concentration of L-cysteine in the fermentation medium would affect the yield of benzenemethanethiol and investigated the effect of different concentrations of L-cysteine on the yield of fermentation products (0.605, 1.21, 1.815, 2.42, 3.025, and 3.63 g/L). As depicted in Figure 5B, the corresponding benzenemethanethiol content level increases with the increase in L-cysteine concentration, reaching the maximum 0.87 mg/L at 3.025 g/L, which is significantly higher than the initial 0.605 g/L addition. However, a decrease in benzenemethanethiol content occurred when the L-cysteine substrate was added above 3.025 g/L. This phenomenon may be attributed to the inhibitory effect of high L-cysteine concentrations on yeast cell viability [43]. Therefore, we selected 3.025 g/L as the optimal amount of L-cysteine for subsequent fermentation experiments.

#### 3.4.3. Effect of Fermentation Temperature

Temperature has a dominant position in the growth and metabolism of microbes [44]. Elevated temperatures can affect enzymes and nucleic acids in microbial strains. Considering the high volatility of benzenemethanethiol, it becomes crucial to carefully control the fermentation temperature within a specific range. Figure 5C demonstrates that the benzenemethanethiol content level in the fermentation broth reached a significant maximum at 28 °C when the fermentation temperature was increased from 20 to 35 °C. This suggests that the carbon–sulfur lyase activity of J14 is highest at 28 °C.

#### 3.4.4. Effect of Fermentation Time

The manufacturing of carbon–sulfur lyase secreted by *Saccharomyces cerevisiae* influences its application in actual fermentation. During the optimization of the baijiu brewing process, the phase of robust carbon–sulfur lyase production plays a crucial role. As depicted in Figure 5D, the carbon–sulfur lyase activity of J14 increased with the extension of fermentation time in the sorghum extract fermentation medium, reaching its maximum at 24 h. Subsequently, the activity declined rapidly, with a slow decrease in enzyme activity after 36 h. Therefore, J14 exhibited optimal carbon–sulfur lyase efficiency after 24 h of fermentation, with a benzenemethanethiol concentration of 0.59 mg/L in the fermentation medium. It is important to note that this value is slightly lower than the previous finding, which could be attributed to the repeated sampling of the fermentation broth during the process and the high volatility of benzenemethanethiol, leading to some loss of the target compound.

## 4. Conclusions

In this study, we identified microorganisms from the baijiu fermentation environment capable of producing benzenemethanethiol. We also discovered that this compound can be produced using L-cysteine and benzaldehyde as precursors under the action of microorganisms or enzymes. Additionally, we isolated 12 yeasts with carbon–sulfur lyase activity from daqu and identified a strain, J14, with high carbon–sulfur lyase activity as belonging to the *Saccharomyces cerevisiae* genus. We determined the production capacity of J14 through the production capacity of benzenemethanethiol. The optimal fermentation conditions were an initial fermentation medium seed volume of 4% (*v/v*), 3.025 g/L L-cysteine addition at 28 °C, and 24 h fermentation. To gain a comprehensive understanding of the complex metabolism of microorganisms, various omics approaches, such as genomics, transcriptomics, proteomics, and metabolomics, can be employed; these techniques enable the study of dynamic processes at the levels of DNA, RNA, proteins, and metabolites, respectively. In future research, whole genome sequencing can be utilized to identify key enzymes involved in benzenemethanethiol production. Additionally, transcriptome analysis coupled with gene knockout and overexpression experiments can help elucidate the metabolic regulatory mechanisms. Such investigations will provide valuable insights into the scientific production of traditional fermented foods and beverages. Overall, this research serves as a valuable reference for enhancing the controlled and efficient manufacturing of traditional fermented products.

## Figures and Tables

**Figure 1 foods-12-02464-f001:**
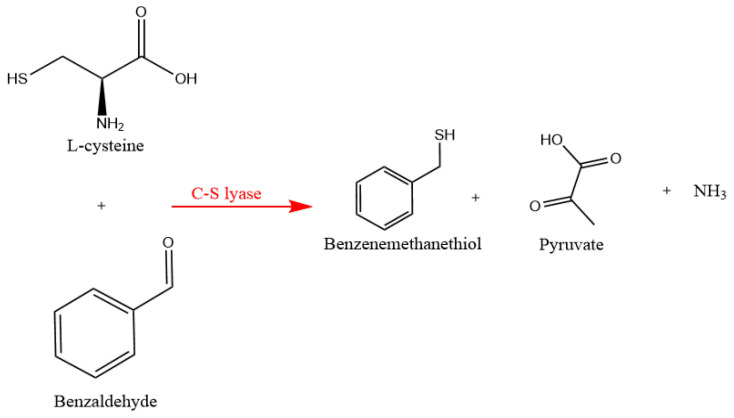
The hypothetical pathway for benzenemethanethiol formation under the action of C-S lyase secreted from yeast. Adapted from [16,19,20,21].

**Figure 2 foods-12-02464-f002:**
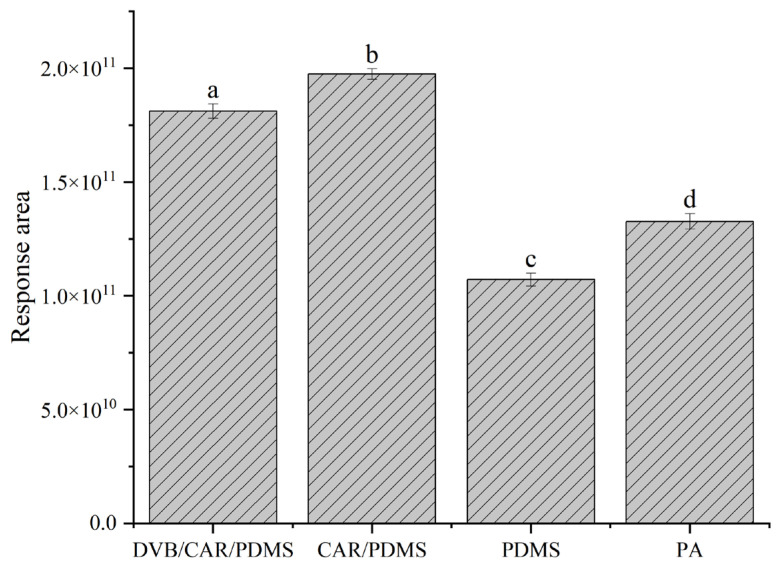
Response areas of benzenemethanethiol extracted with different fibers measured in the sorghum extract medium. Values not sharing the same superscript letter (a, b, c, d) on the top bar are different according to Duncan’s test.

**Figure 3 foods-12-02464-f003:**
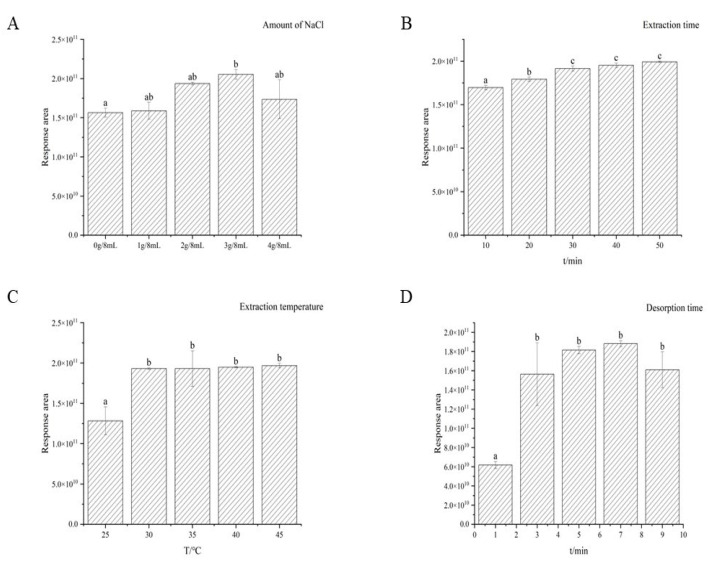
Response areas of benzenemethanethiol extracted under different conditions measured in the sorghum extract medium. Among which, (**A**): The influence of different sodium chloride addition levels on extraction efficiency; (**B**): The influence of different extraction times on extraction efficiency; (**C**): The influence of different extraction temperatures on extraction efficiency; (**D**): The influence of different desorption times on extraction efficiency. Values not sharing the same superscript letter (a, b, c) on the top bar are different based on Duncan’s test.

**Figure 4 foods-12-02464-f004:**
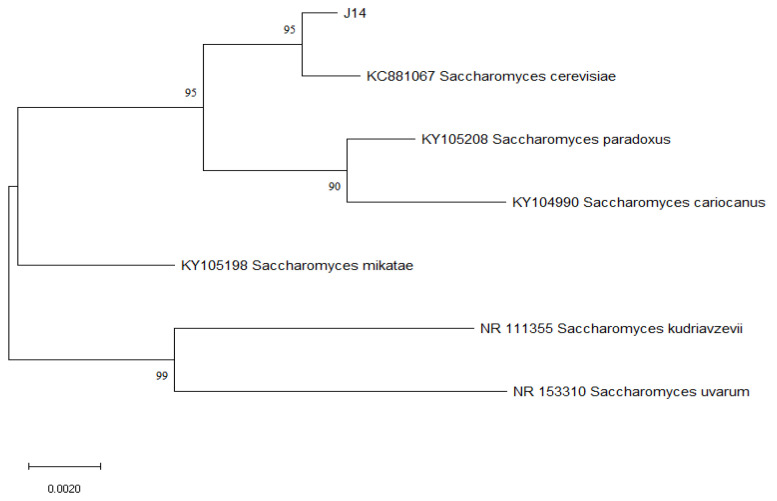
Analysis of the evolutionary relationship of strain J14 with other strains using MEGA software v11 with the neighbor-joining method and a bootstrap value of 1000.

**Figure 5 foods-12-02464-f005:**
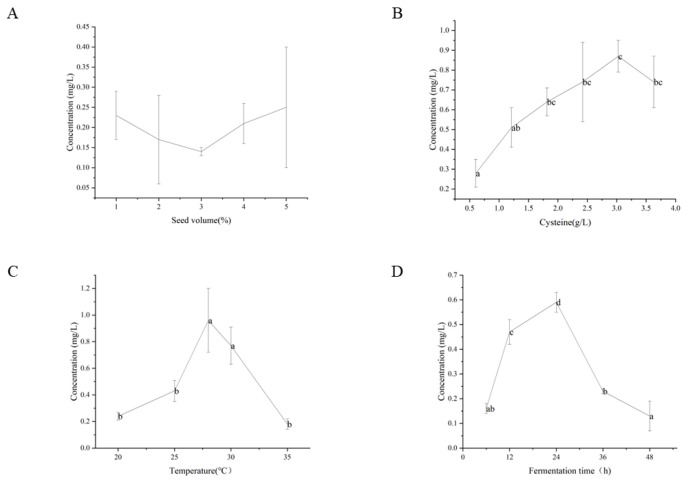
Influence of fermentation factors on the yield of benzenemethanethiol: (**A**) effect of seed volume; (**B**) effect of L-cysteine supplementation amount; (**C**) effect of fermentation temperature; (**D**) effect of fermentation time. Values not sharing the same superscript letter (a, b, c, d) on the top bar are different according to Duncan’s test.

**Table 1 foods-12-02464-t001:** Main instruments.

Instrument	Manufacturer
PCR instrument (MG96+)	LongGene Scientific Instruments, Inc. (Hangzhou, China)
Gel imager (JY04S-3C)	Junyi Electrophoresis Co., Ltd. (Beijing, China)
JY300C ligand electrophoresis system	Junyi Electrophoresis Co., Ltd. (Beijing, China)
Medical centrifuge (H1650-W)	Hunan Xiangyi Co., Ltd. (Hunan, China)
3730XL sequencer (625-0020)	Applied Biosystems (Waltham, MA, USA)

**Table 2 foods-12-02464-t002:** PCR reaction system.

Reactant	Volume
10 × Ex Taq buffer	5.0 μL
2.5 mM dNTP Mix	4.0 μL
10p Primer 1	2.0 μL
10p Primer 2	2.0 μL
5u Ex Taq	0.5 μL
Template	2.0 μL
ddH_2_O	34.5 μL
Total volume	50 μL

**Table 3 foods-12-02464-t003:** The PCR reaction conditions.

	Temperature (°C)	Time	Cycles
Initial denaturation	94	3 min	1
Denaturation	94	30 s	24
Primer annealing	54	30 s	
Primer extension	72	1 min 30 s	
Final extension	72	10 min	1

**Table 4 foods-12-02464-t004:** Preliminary screening results of yeasts.

Name	Concentration of BM (mg/L)	Name	Concentration of BM (mg/L)
J2 ^a^	0.12 ± 0.04	J14 ^d^	0.50 ± 0.07
J5 ^bcd^	0.34 ± 0.16	J15 ^bcd^	0.33 ± 0.04
J6 ^abc^	0.27 ± 0.11	J19 ^a^	0.13 ± 0.04
J9 ^bcd^	0.33 ± 0.04	J21 ^a^	0.11 ± 0.02
J10 ^ab^	0.21 ± 0.06	J23 ^cd^	0.43 ± 0.12
J13 ^cd^	0.41 ± 0.08	J27 ^ab^	0.20 ± 0.03

BM—benzenemethanethiol. Values not sharing the same superscript letter (a, b, c, d) on the top bar are different according to Duncan’s test.

## Data Availability

The data are available from the corresponding author.

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
