# Peer review of "Isolation and Characterization of Yeast with Benzenemethanethiol Synthesis Ability Isolated from Baijiu Daqu"

_foods, 2023, doi:10.3390/foods12132464_

Round 1

Reviewer 1 Report

The topic of the manuscript is interesting and a lot of experimental work was done. However, there are a lot of things that should be corrected in order to improve manuscript quality.

Title

The title doesn't correspond with the manuscript topic. The half of paper is for isolation and identification of yeast strains but the other is for the optimisation of conditions for baijiu fermentation with this strain.

Introduction

It will be useful to write more about Baijiu technology because as I have understood it is common drink for China and is unknown for all other countries.

ln 44 percent ->%

Material and methods

ln 111 Capital letter at the beginning of sentences

ln 118 10-1-10-6 ->10-1-10-6

Results and discussion

ln 222-224 You have to delete it

ln 383-386 Unclear sentence

Author Response

Thank you for your careful reading and valuable suggestions. Your professional opinions have played a crucial role in our research. We will seriously consider and adopt your suggestions. Once again, thank you for your support and guidance.

Reviewer 2 Report

The authors did an interesting study and they showed that native isolated yeasts are capable of producing benzenemethanethiol from a baijiu fermentation environment. They also validated that benzenemethanethiol can be produced through L cysteine and benzaldehyde as precursors under the action of microorganisms/enzymes. The article is well-written and well-structured. I have just a few suggesting for the text improvement which are inserted directly in the PDF.

Author Response

We are extremely grateful for the valuable opinions and suggestions provided by the reviewers on our research. Your professional evaluation has been of great assistance to our study. We will take your suggestions seriously and continuously improve the quality of our research. Once again, thank you for your support and help.

Reviewer 3 Report

Well constructed article. For better scientific quality of work, all isolated strains should be identified. Please put a photo of the agarose gel.

Author Response

We would like to express our sincerest gratitude and respect to the reviewers for their careful review and valuable suggestions on our research. Your professional opinions have significant importance to our study. We have taken your suggestions seriously and continuously improved the quality of our research. Once again, thank you for your support and guidance.

Reviewer 4 Report

While the topic of the paper is worthy of investigation, the authors should clarify some issues related to statistic:

a) why and how optimization by single factor was conducted? Why not an experimental design? The use of a single factor protocol could lead to systematic bias, as variables interact.

b) how many technical replicates and how many independent batches

Author Response

Dear Reviewer,

I would like to express my sincere gratitude for your time and effort in reviewing my manuscript. Your insightful comments and constructive feedback have been invaluable to the improvement of my research. Your expertise and attention to detail have greatly contributed to the quality of the final manuscript.

I appreciate the time you have taken to carefully review my work and provide thoughtful feedback. Your comments have helped me to better understand the strengths and weaknesses of my research, and have enabled me to make the necessary revisions to improve the manuscript.

Once again, thank you for your time and effort in reviewing my manuscript. Your feedback has been instrumental in the success of my research, and I am grateful for your support.

Round 2

Reviewer 1 Report

Dear Authors,

As I have written in previous comment you have done a lot of experimental work. However, I still don't understand how baijiu is produced. As I told you last time this beverage is familiar only to people in China or may be in other regions of Asia but not in other continents, so be more detailed of its production. Moreover, what is daqu from which you have isolated yeast?

The title sounds much better but I'm not sure that characterization is the correct word

Author Response

I would like to express my heartfelt gratitude to the reviewer for the invaluable feedback and constructive comments. I have adopted your suggestions and made the necessary revisions.

Reviewer 4 Report

The authors addressed my issues; I think that the clarification for the single factor optimization reported in the rebuttal letters should be added in Materials and Methods 

Author Response

Thank you very much for your valuable suggestions. We have adopted your advice and made the necessary modifications
